# Cervical *Staphylococcus aureus* Infection after Receiving the Third Dose of COVID-19 Vaccination: A Case Report

**DOI:** 10.3390/vaccines10081276

**Published:** 2022-08-08

**Authors:** Tao Li, Hongqi Zhang, Yuxiang Wang, Yunjia Wang, Qile Gao, Mingxing Tang, Shaohua Liu, Gengming Zhang, Chaofeng Guo

**Affiliations:** 1Department of Spine Surgery and Orthopaedics, Xiangya Hospital, Central South University, Changsha 410008, China; 2National Clinical Research Center for Geriatric Disorders, Xiangya Hospital, Central South University, Changsha 410008, China

**Keywords:** *Staphylococcus aureus*, infection, COVID-19, vaccine, cervical vertebra

## Abstract

Introduction: Vaccination is one of the most effective ways to control the COVID-19 pandemic. However, as the number of people vaccinated against COVID-19 continues to increase, there are more reports on the safety of vaccines. So far, there have been no reported cases of spinal infection associated with COVID-19 vaccination. Recently, we admitted a patient who developed cervical *Staphylococcus aureus* infection resulting in high paraplegia after receiving the third dose of COVID-19 vaccine when the symptoms of cold did not completely disappear. Case presentation: The patient was a 70-year-old man who received the third injection of COVID-19 vaccine when the cold symptoms were not completely gone. On the day after the injection, the patient developed severe neck and shoulder pain, accompanied by numbness and fatigue in the limbs. MRI examination of the cervical spine on day 6 after vaccination showed no obvious signs of infection. The patient had progressive weakness in the extremities. On the ninth day after vaccination, the patient developed paralysis of both lower limbs and significant sensory loss. Cervical abscess and cervical spinal cord injury were considered for cervical CT and MRI examination on the 15th day after vaccination. We used an anterior approach to remove as much of the lesion as possible. *Staphylococcus aureus* was detected and antibiotic treatment was continued after surgery. The patient’s pain symptoms were significantly relieved, which prevented the abscess from further pressing the spinal cord and provided possible conditions for the recovery of neurological function in the later stage. Conclusion: This case is the first reported cervical *Staphylococcus aureus* infection resulting in high paraplegia after receiving the third dose of COVID-19 vaccine with low immunity. This case raises awareness of this rare but potentially life-threatening adverse reaction, and reminds people to hold off when their immune system is weakened.

## 1. Introduction

COVID-19 swept the world in 2019, posing a great threat to human health and becoming a new public health issue in the 21st century [1]. More recently, vaccination has provided an effective way to control the COVID-19 pandemic. As the number of people vaccinated against COVID-19 increases, so do studies on the safety of the vaccine [2]. Common side effects after vaccination are mild local reactions such as redness, swelling, pain, and skin necrosis [3]. However, some people will have different degrees of adverse or even life-threatening reactions after vaccination [4]. For example, acute transverse myelitis and Guillain–Barre syndrome after COVID-19 vaccination have been reported in the literature [5]. However, there have been no reported cases of spinal infection associated with COVID-19 vaccination. We recently admitted a patient who developed cervical *Staphylococcus aureus* infection after receiving the third dose of Sinovac COVID-19 vaccine (Beijing, China) when cold symptoms were not completely gone, as reported below.

## 2. Case Presentation

The patient was a 70-year-old male patient with a history of hypertension who was transferred to the emergency department of our hospital on 5 April 2022 (the 13th day after vaccination, denoted as “D13”) due to neck and shoulder pain accompanied by numbness and fatigue for 12 days. He was admitted to the intensive care unit (ICU) of our hospital on 9 April. The patient had the cold symptoms of nasal stuffiness and discharge, sneezing, sore throat, and cough before vaccination, and nasal congestion symptoms on the day of vaccination, but no other symptoms. The patient suddenly developed severe neck and shoulder pain 12 days after receiving the third dose of COVID-19 vaccine while cold symptoms were not completely gone on 23 March 2022. He went to a local hospital and was diagnosed with “periarthritis of shoulder”. In the evening, he began to develop weakness and numbness of limbs, accompanied by mobility disorder and inability to walk, accompanied by headache and dizziness, without nausea and vomiting. Symptomatic treatment was given, but the symptoms did not alleviate significantly and became progressively worse, so he was transferred to a local superior hospital. Laboratory blood samples taken on 29 March 2022 (6 days after vaccination, similarly denoted as “D6”) showed amounts of white blood cells (WBC) at 12.1 × 10^9^/L, neutrophil count percent (NEUT%) at 89.2%, procalcitonin (PCT) at 0.45 ng/mL, C-reactive protein (CRP) at 100.0 mg/L, and erythrocyte sedimentation rate (ESR) at 53 mm/h (Table 1). Cervical X-ray at 6 days after vaccination (D6) showed straightening of cervical curvature and hyperosteogeny (Figure 1a). Plain CT scan of cervical spine showed hyperosteogeny of cervical spine and slight herniation of C3/4, C4/5, and C5/6 intervertebral discs (Figure 1b). Cervical MRI plain scan showed cervical instability, spinal cord degeneration at C3 to C6 vertebral body level, backward disc herniation at C3/4, C4/5, C5/6, and C6/7, and spinal canal stenosis at C6/7 level (Figure 1c). During the hospitalization, the weakness of the limbs gradually worsened, and the patient developed paralysis of the lower extremities and hypoesthesia on 1 April 2022 (D9). Fever occurred once, with a body temperature of 38.6 ℃ accompanied by chills, and there was no fever since then. The diagnosis of the local hospital considered the possibility of cervical vertebra infection, and meropenem and vancomycin were combined to fight infection, but the symptoms were not significantly relieved. For further treatment, the patient was transferred to the emergency department of our hospital. The patient developed spontaneous illness with poor energy, appetite, and sleep. Physical examination: low breath sound in both lungs, no obvious subcutaneous mass in neck, and no sinus or ulceration in skin. The physiological lordosis of the cervical spine disappeared and the movement was significantly limited. There was obvious tenderness in spinous processes and paraspinal muscles from C3 to C6, radiating pain in both upper arms, shoulders and back. The sensation below the line plane of both nipples decreased significantly. Muscle strength: left upper limb is grade 1, right upper limb is grade 1+, lower limb is grade 0. Muscle tension: both upper limbs decreased significantly, both lower limbs disappeared. Bilateral Chaddock signs were positive, but both cervical resistance and bilateral Babinski signs were negative. Laboratory blood samples taken on 5 April 2022 (D13) showed amounts of WBC at 14.1 × 10^9^/L, NEUT% at 85.5%, PCT at 0.16 ng/mL, and ESR at 67 mm/h (Table 1). Imaging examination: Cervical X-ray films at 15 days after vaccination (D15) in our hospital suggested possible intervertebral lesions of C5/6 and C6/7 (Figure 1d). Plain CT scan of the cervical spine and lung revealed increased density in the right margin of the C5 and C6 vertebrae: nature undetermined, C2/3 and C3/4 discs slightly protruded backwards (Figure 1e). MRI plain scan of cervical spine reveals abnormal signals in the spinal cord at the level of C3 to C6 vertebrae, compressive changes, and abnormal signals in C5 vertebrae, abnormal signals and flattening in C5/6 intervertebral discs: infectious lesions? (Figure 1f). Cervical abscess, cervical spinal cord injury, and paraplegia were considered for cervical CT and MRI examination. The patient had cervical abscess compression on the spinal cord and was indicated for surgery. He was temporarily given anti-infection treatment of imipenem, cilastatin, and levofloxacin, and was transferred to ICU for intensive care and waiting for surgery. Laboratory test results on 9 April 2022 (D17) showed amounts of WBC at 7.8 × 10^9^/L, NEUT% at 74.9%, PCT at 0.30 ng/mL, and CRP at 93.3 mg/L (Table 1). The patient did not have symptoms such as low fever and night sweats, and all indicators related to tuberculosis infection were negative, so the possibility of tuberculosis infection was considered to be low. The COVID-19 nucleic acid tests were negative before the third vaccination and during hospitalization. Combined with the patient’s symptoms, signs, and auxiliary examination results, the possibility of suppurative infection is considered to be high for cervical bone destruction at present.

Imaging suggested cervical vertebra infection, and inflammatory indicators such as WBC, PCT, CRP, and ESR were all high. Levofloxacin and linezolid were given preoperatively to fight infection. On 11 April 2022 (D19), the patient was treated by anterior debridement, decompression, bone grafting, fusion, and instrumentation under general anesthesia. During the operation, the prevertebral fascia was incised, which found hyperplasia, thickening, edema, and adhesion to the anterior longitudinal ligament. Intraoperative intervertebral disc damage was observed at C3/4, C4/5, and C5/6, and organic abscess tissues in the anterior cervical vertebra, C5/6 intervertebral space and posterior spinal canal. The lesions were carefully removed and sent for pathological examination, culture and metagenomic next-generation sequencing (mNGS). Anti-infection regimen of levofloxacin and linezolid was continued after surgery. *Staphylococcus aureus* detected by mNGS (sequence number 206). No bacteria were found in anaerobic and aerobic culture of bone tissue after operation. Pathological findings suggested chronic suppuration of C5/6 intervertebral disc with dead bone formation (Appendix A). At present, cervical *Staphylococcus aureus* infection has been confirmed. In order to find the source of pathogen, sputum smear, and culture were performed again, but no *Staphylococcus aureus* was found. On 12 April 2022 (1st postoperative day, marked as “S1”), the re-examination index showed amounts of WBC at 8.3 × 10^9^/L, NEUT% at 90.4%, ESR at 41 mm/h, CRP at 49.2 mg/L, and PCT at 0.14 ng/mL (Table 1). Re-examination of X-ray and CT showed that the internal fixation device was in good position without loosening, fracture, prolapse, or displacement (Figure 1g,h).

The patient was treated in the ICU of our hospital for 4 days after surgery, and his general condition improved, neck and shoulder pain relieved, with no fever. On 15 April 2022 (the fourth day after surgery, similarly marked as “S4”), the re-examination index showed amounts of WBC at 7.4 × 10^9^/L, NEUT% at 75.6%, CRP at 35.3 mg/L, and PCT was normal (Table 1). The patient was transferred to a rehabilitation hospital for further rehabilitation and anti-infective treatment.

## 3. Discussion

According to literature review, cervical *Staphylococcus aureus* infection in this case is the first report related to COVID-19 vaccination. The pathogen of COVID-19 is a single-stranded plus stranded RNA virus in the coronavirus family, originally named “2019-nCoV” and later changed to “SARS-CoV-2”. COVID-19 can widely infect hosts and cause serious diseases such as the common cold, middle east respiratory syndrome (MERS), and severe acute respiratory syndrome (SARS), which seriously endanger human health and safety [2]. Vaccination is considered the most effective way to prevent COVID-19, significantly reducing the risk of contracting the disease in a population. However, the development and safety of COVID-19 vaccines have attracted worldwide attention. Any health-related problems that occur after vaccination are considered a post-vaccination adverse event (PVAE) [6]. PVAE may be related to the vaccine itself (called a side effect) or may be an occasional event that occurs after vaccination. In this case, the patient received the third injection of COVID-19 vaccine before the cold symptoms completely disappeared. MRI examination results on the 15th day after vaccination showed cervical vertebra infection, resulting in cervical abscess compression on the cervical spinal cord (Figure 1f), resulting in high paraplegia, which was considered to be an accidental event related to COVID-19 vaccine inoculation.

In recent years, the incidence of infectious spondylitis is increasing and diagnosis is difficult [7,8,9]. Gasbarrini et al. reported an average delay of 2 to 4 months in diagnosing spinal infection [10]. Babic et al. believe that despite the availability of advanced instruments, there are still diagnostic challenges and therefore recommend the need to maintain a high level of vigilance for spinal infection in patients who are more susceptible to infection (e.g., immunocompromised patients, diabetics, patients receiving spinal injections) [11]. *Staphylococcus aureus* is the most common pathogen of spinal infection and the most common route of transmission is blood-borne [7,8]. The source of *Staphylococcus aureus* in this case is unknown. It is possible that a staff member involved in vaccinations had a *Staphylococcus aureus* infection either on a skin surface or intranasally. We believe that the vaccination with COVID-19 vaccine when the patient’s immune system was low reduced the immune capacity of the body to protect against pathogens, resulting in the occurrence of infection and the formation of abscess pressing the spinal cord. The reasons are as follows: 1. Patients were vaccinated when their cold symptoms did not completely disappear; 2. MRI scan at the early stage of vaccination (D6, Figure 1c) did not show conclusive evidence of cervical infection, but re-examination MRI (D15, Figure 1f) showed cervical infection and spinal abscess compression, and organic abscess tissue in the anterior cervical vertebra, C5/6 intervertebral space and posterior spinal canal could be observed during the operation. 3. No history of invasive cervical examination or treatment, and no infection found in other parts of the body. 4. The patient only had a history of hypertension and denied any other medical history. No history of taking immunosuppressive drugs or hormones three months before onset. In addition, the patient received three doses of vaccines from different manufacturers, including the first dose of Sinopharm vaccine (WIBP, Wuhan, China) on 21 July 2021, the second dose of Sinopharm vaccine (BIBP, Beijing, China) on 18 August 2021.

The best treatment for spinal infection complicated by abscess formation includes surgical decompression and drainage, requiring a full course of antibiotics [12,13]. In this case, the patient had cervical vertebra infection combined with prevertebral space abscess, and there were paralysis symptoms caused by the abscess compressing the spinal cord. Therefore, surgical treatment was performed early to remove the lesion and relieve the abscess compression as soon as possible. In addition, in this case, the abscess was mainly located in the front of cervical vertebra and intervertebral space, so we adopted anterior surgery and removed the lesion as much as possible. Antibiotics were given after surgery, which significantly relieved the patient’s pain symptoms and prevented the abscess from further compressing the spinal cord, providing possible conditions for the recovery of neurological function in the later stage.

Outbreaks of cervical *Staphylococcus aureus* infection after vaccination are very rare. The case reminds people to delay vaccination when their immune system is weakened.

## 4. Conclusions

To our knowledge, the case is the first report of cervical *Staphylococcus aureus* infection, observed shortly after the third dose of COVID-19 vaccine, described in the scientific literature. This is a 70-year-old patient who developed high paraplegia due to cervical *Staphylococcus aureus* infection after receiving the third dose of COVID-19 vaccine. The case raises awareness of a rare but potentially life-threatening adverse reaction, prompting people to delay vaccination if their immune system is compromised.

## Figures and Tables

**Figure 1 vaccines-10-01276-f001:**
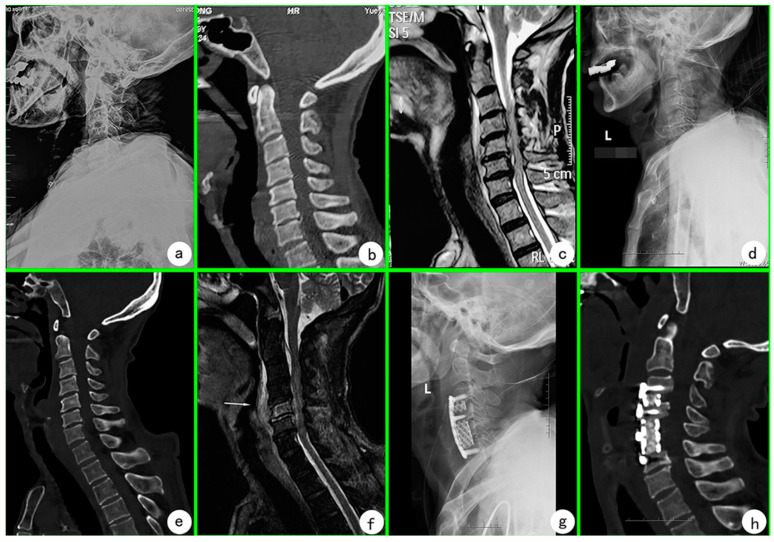
A 70-year-old male patient was admitted with neck and shoulder pain accompanied by numbness and fatigue for 12 days. (**a**) Cervical X-ray at 6 days after vaccination (D6) showed straightening of cervical curvature and hyperosteogeny. (**b**) On D6, plain CT scan of cervical spine showed hyperosteogeny of cervical spine and slight herniation of C3/4, C4/5, and C5/6 intervertebral discs. (**c**) On D6, cervical MRI plain scan showed cervical instability, spinal cord degeneration at C3 to C6 vertebral body level, backward disc herniation at C3/4, C4/5, C5/6, and C6/7, and spinal canal stenosis at C6/7 level. (**d**) Cervical X-ray at 15 days after vaccination (D15) showed possible intervertebral lesions of C5/6 and C6/7. (**e**) On D15, CT scan of cervical spine showed increased density in the right margin of C5 and C6 vertebrae: nature undetermined, C2/3 and C3/4 disc slightly protruded backwards. (**f**) On D15, MRI plain scan of cervical spine showed abnormal signals of spinal cord at the level of C3 to C6 vertebrae, compressive changes of C5 vertebrae and abnormal signals and flattening of C5/6 intervertebral disc. The cause is to be investigated: infectious lesions. (**g**,**h**) Postoperative cervical X-ray and CT plain scan showed normal physiological lordosis of cervical spine, good position of internal fixation device, no loosening, fracture, prolapse or displacement.

**Table 1 vaccines-10-01276-t001:** Trend of the WBC and Inflammatory indicators.

Date	WBC (×10^9^/L)	NEUT%	CRP (mg/L)	ESR (mm/h)	PCT (ng/mL)
29 March 2022 (D6)	12.1	89.2	100.0	53	0.45
5 April 2022 (D13)	14.1	85.5	-	67	0.16
9 April 2022 (D17)	7.8	74.9	93.3	-	0.30
12 April 2022 (S1)	8.3	90.4	49.2	41	0.14
15 April 2022 (S4)	7.4	75.6	35.3	-	<0.05

Note: “D6” stands for day 6 after vaccination, “S1” stands for day 1 after surgery, and so on. Normal ranges: WBC: (4–10) × 10^9^/L, NEUT%: 40–75%, CRP: 0–8.0 mg/L, ESR: 0–21 mm/h, PCT: ≤0.1 ng/mL.

## Data Availability

The original contributions presented in the study are included in the article/Appendix A, further inquiries can be directed to the corresponding author.

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
