# Peer review of "Cervical Staphylococcus aureus Infection after Receiving the Third Dose of COVID-19 Vaccination: A Case Report"

_vaccines, 2022, doi:10.3390/vaccines10081276_

Round 1
Reviewer 1 Report
Thank you for this meticulously reported case.
The key question is how the patient acquired the infection. You commented that the patient had “no sinus or ulceration in skin.” Please examine the culture reports of your regional laboratory system to see if there has been an increase in staphylococcus aureus infections. Please also check if other COVID vaccinated patients had an increase in infections during this period. It is possible that a staff member involved in vaccinations had a staphylococcus aureus infection either on a skin surface or intranasally.
English text is excellent. Please check:
15 the symptoms of cold did not completely disappear [you mention cold symptoms several times. [Please define what you mean by “cold” and your diagnostic criteria for “cold” symptoms]
54 12 days ago after receiving [please delete ago]
the symptoms did not relieve significantly [please change to “were not significantly relieved”]
131 organic purpus tissues [meaning? please amend purpus]
137 suppuratitis [please correct to suppuration]
Reviewer 2 Report
It is important for people to know the different reverse effects of COVID-19 vaccination. Li et al report the first severe cervical infection caused by Staphylococcus aureus after COVID-19 vaccination. After reading this manuscript, I have some questions as follows:
Line 1-3, suggested title:
Cervical staphylococcus aureus infection after receiving the third dose of COVID-19 Vaccination: a case report . For the reason that the authors could not conclude from the evidences in this case to relate the S. aureus infection to COVID-19 vaccine.
Line 11-36,
Could you shorten the abstract by reducing the details? It is important to point out why this case is essential for publication. Please use two or three sentences to describe how special your case is and what the related cases reported are until now.
Line 45,
There are related cases of transverse myelitis, which should be mentioned here in the background.
Line 49,
Considering that the patient is an old man, the authors should describe here his past medical history.
Line 54,
Could you provide the information of three times vaccination? The time, the manufacturer, and the type of vaccine.
Line 110-144,
Although RT-PCR could not exclude the possibility of COVID-19 infection, the result should also be mentioned in this study considering it is related to COVID-19 vaccine and infection disease. It might also be a complication of COVID-19 infection.
Line 206,
There is not enough evidence in this case to make such a conclusion about different manufacturers.
Line 207,
In order to not add public fear of the vaccine, the author should mention in the conclusion part that developing serious complications is of very rare chance while the many benefits of COVID-19 vaccines should be emphasized.
Scheme 1,
The author should describe what color and shape represents the Staphylococcus aureus for the convenience of the readers.
Round 2
Reviewer 2 Report
Thank you for the answers and modifications to your manuscript. I have some more questions considering your modification.
Line 105-107, 'The patient's COVID-19 nucleic acid test was negative two weeks before vaccination and during hospitalization, so COVID-19 infection is not supported.' What do you mean by this sentence? Do you mean that COVID-19 was not detected either before vaccination or during hospitalization? If so, you could write: 'The COVID-19 nucleic acid tests were negative before the third vaccination and during hospitalization. '
Line 180-182, do not add the information on the shape and color of S. aureus here. It is not relevant in this place. What I meant is that you should point out what is S. aureus in your figure 'Scheme 1' (Line 235-237).
Line 193-199, you can not make the conclusion that vaccines from different manufacturers matter. So, please do not make this kind of conclusion in your manuscript. It will not help you and just reflect the authors' lack of logical thinking.
Author Response
Thank you for your review again.
Point 1:
Line 105-107, 'The patient's COVID-19 nucleic acid test was negative two weeks before vaccination and during hospitalization, so COVID-19 infection is not supported.' What do you mean by this sentence? Do you mean that COVID-19 was not detected either before vaccination or during hospitalization? If so, you could write: 'The COVID-19 nucleic acid tests were negative before the third vaccination and during hospitalization. '
Response 1: Thank you for your constructive comments. We have changed to "The COVID-19 nucleic acid tests were negative before the third vaccination and during hospitalization. " (Line 105-107).
Point 2:
Line 180-182, do not add the information on the shape and color of S. aureus here. It is not relevant in this place. What I meant is that you should point out what is S. aureus in your figure 'Scheme 1' (Line 235-237).
Response 2: Sorry, I misunderstood. Thank you for your constructive comments again. We have point out what is S. aureus in the figure 'Supplementary Figure 1' (Line 235-237).
Point 3:
Line 193-199, you can not make the conclusion that vaccines from different manufacturers matter. So, please do not make this kind of conclusion in your manuscript. It will not help you and just reflect the authors' lack of logical thinking.
Response 3: Thank you again for your valuable advice. We have deleted this part of the conclusion in the manuscript.